# EMBEDDING-CONVERTER: A UNIFIED FRAMEWORK FOR CROSS-MODEL EMBEDDING TRANSFORMATION

## ABSTRACT

Embeddings, numerical representations of data like text and images, are fundamental to machine learning. However, the continuous emergence of new embedding models poses a challenge: migrating to these potentially superior models often requires computationally expensive re-embedding of entire datasets even without guarantees of improvement. This paper introduces Embedding-Converter, a unified framework and a novel paradigm for efficiently converting embeddings between different models, eliminating the need for costly re-embedding. In real-world scenarios, the proposed method yields O(100) times faster and cheaper computation of embeddings with new models. Our experiments demonstrate that Embedding-Converter not only facilitates seamless transitions to new models but can even surpass the source model's performance, approaching that of the target model. This enables efficient evaluation of new embedding models and promotes wider adoption by reducing the overhead associated with model switching. Moreover, Embedding-Converter addresses latency constraints by enabling the use of smaller models for online tasks while leveraging larger models for offline processing. By encouraging users to release converters alongside new embedding models, Embedding-Converter fosters a more dynamic and user-friendly paradigm for embedding model development and deployment.

## 1 INTRODUCTION

Embeddings are the cornerstone of many machine learning systems. They transform complex data, such as text and images, into a format readily processed by computers: numerical vectors. These vectorized representations serve as the foundation for a wide range of applications, including search, clustering, anomaly detection, classification, and information retrieval (Wang et al., 2016; Huang et al., 2020; Zhai et al., 2019).

However, the landscape of embedding models is becoming increasingly complex. A multitude of models are available, each with its own strengths and weaknesses (Wang et al., 2022; Li et al., 2023; Lee et al., 2024a). This diversity, while offering flexibility, presents a significant challenge: determining the optimal embedding model for a specific task often necessitates a computationally expensive and time-consuming evaluation process, especially when dealing with massive datasets. Consider the scenario of selecting the best embedding model for a billion text passages. Evaluating each candidate model requires generating embeddings for all billion passages, a daunting computational undertaking (see Appendix B for detailed computational complexities). This challenge is further exacerbated by the continuous emergence of new and improved models, forcing a repetitive cycle of re-embedding with no guarantee of substantial performance gains. Furthermore, the lack of compatibility between different embedding models, even within the same family (e.g., Google's Gecko (Lee et al., 2024b) or OpenAI's embeddings (Neelakantan et al., 2022)), poses a significant obstacle. This incompatibility necessitates a complete recomputation of embeddings whenever a user wishes to explore a new model or upgrade to a newer version, hindering efficient experimentation and adoption of state-of-the-art techniques. This laborious process presents a major roadblock to leveraging the latest advancements in embedding models for real-world applications.

To address the aforementioned challenges, this paper introduces Embedding-Converter, a unified framework designed to facilitate seamless transitions between different embedding models. Functioning as a universal translator for embedding spaces, Embedding-Converter empowers machine

(a) Conventional evaluation framework    (b) Proposed evaluation framework

Figure 1: Illustrating the efficiency benefits of using Embedding-Converter when the scenario of comparing embedding models A and B. Embedding models A and B can represent different versions of the same model or entirely distinct models. (Left) Conventional approach for evaluating of a new embedding model (B) requires re-embedding the entire corpus, and it incurs significant computational cost. (Right) Embedding-Converter efficiently transforms existing embeddings from embedding model A to embedding model B, dramatically reducing the computational overhead.

learning practitioners to effortlessly explore new models, upgrade to the latest versions, and even switch between entirely different model families (e.g., Google Gecko vs. OpenAI) without incurring the computational cost of re-embedding their data. This capability accelerates the adoption of new technologies and provides greater flexibility in managing embedding models (see Fig. 1).

Developing such a converter presents unique challenges. It requires learning an efficient mapping between potentially disparate high-dimensional spaces from unlabeled text data (see Fig. 2(b)). The model must possess sufficient capacity to enable effective transfer while avoiding overfitting, and the training process must be guided by appropriate loss functions to ensure accurate conversion. This paper elucidates the novel methodological approaches employed in the development of Embedding-Converter to address these challenges. Through extensive experiments across diverse scenarios, we demonstrate its efficacy and provide insights into its key components. Our evaluation encompasses various conversion scenarios, including intra-model conversions (between different versions within the same model family), inter-model conversions, and conversions between models with different embedding dimensions. Furthermore, we assess the performance of Embedding-Converter on a range of downstream tasks involving embeddings, such as retrieval and semantic textual similarity.

Our experiments consistently demonstrate that the converted embeddings closely resemble the target embeddings, effectively surpassing source model performances on downstream tasks. The main contributions of this paper can be summarized as follows:

- We introduce Embedding-Converter, a unified framework that enables cost-effective conversion between different embedding models. Embedding-Converter significantly reduces the computational overhead associated with migrating from one embedding model to another, facilitating efficient exploration and adoption of new models (more than 100x reductions in terms of both computation cost and time).

- Through comprehensive empirical evaluation, we demonstrate that Embedding-Converter effectively surpasses the performance of source embedding models on various downstream tasks, ensuring marginal accuracy degradation from target models during conversion.

## 2 RELATED WORK

### 2.1 EMBEDDING MODELS

Embedding models have become indispensable tools for a wide range of applications, including information retrieval, search, and various other downstream tasks. Driven by the pursuit of improved performance, the field of embedding models is rapidly evolving, with industry leaders such as OpenAI (Neelakantan et al., 2022) and Google (Lee et al., 2024b) continuously releasing new and improved versions. This trend is further exemplified by the competitive landscape of the MTEB leaderboard (Muennighoff et al., 2022), where industrial models like NV-Embed (Lee et al., 2024a),

SFR-Embedding (Rui Meng, 2024), and GTE-Qwen (Li et al., 2023) frequently update their versions to achieve top rankings. Furthermore, there are continual contributions from academia to this vibrant ecosystem with models like General Text Embedding (GTE) (Li et al., 2023) and Generalizable T5-based dense Retrievers (GTR) (Ni et al., 2021), while multimodal embeddings are represented by models like CLIP (Radford et al., 2021) and CoCA (Yu et al., 2022). However, this rapid progress and diversity come at a cost – a lack of compatibility between different embedding models, even across versions within the same family. As evidenced by the varying performance rankings across datasets in the MTEB leaderboard, identifying the optimal embedding model for a specific task or dataset often necessitates evaluating multiple models. This process can be computationally expensive and time-consuming, especially for large corpora, due to the need for re-embedding the entire dataset with each new model. This paper introduces a unified framework to address this challenge. We propose an efficient Embedding-Converter that enables seamless transitions between different embedding models without requiring recomputation of the entire embedding space. This tool empowers machine learning practitioners to readily evaluate various models on their datasets and facilitates effortless migration between models, fostering greater flexibility and efficiency in the development and deployment of embedding-based applications.

## 2.2 Vector space transformation

The task of converting embeddings between different models can be framed as a vector space transformation problem, where the goal is to map numerical vectors from one vector space to another. This is a classic problem in linear algebra with various established approaches, including linear transformations (Marcus, 1971), change of basis (Shores et al., 2007), and kernel methods (Treves, 2013). However, these techniques often assume that the target vector spaces are not predefined, which is not the case with pre-trained embedding models.

Existing research on cross-lingual embedding mapping, such as the work by (Artetxe et al., 2017) and (Conneau et al., 2017), explores techniques for aligning word embedding spaces across different languages. These methods, while relevant, primarily focus on word-level embeddings and might not be directly applicable to embeddings for longer text. Domain adaptation is another related area that investigates adapting embeddings from a source domain to a target domain. (Wang et al., 2021) and (Schopf et al., 2023) propose methods for domain adaptation in embedding spaces, while (Yoon et al., 2024) explore customizing pre-trained embeddings with labeled data. However, these approaches are often tailored to specific domain adaptation scenarios. In contrast, the Embedding-Converter proposed in this paper offers a more versatile solution, capable of converting any sentence embedding from one model to another, regardless of the specific domain or task. This general-purpose applicability distinguishes our approach from prior works and broadens its potential impact across various embedding-based applications.

While some research explores model compatibility in the image domain, these approaches differ significantly from ours. Methods like Backward Compatible Training (BCT) Shen et al. (2020); Hu et al. (2022) require modifying the training process of new models, which is infeasible in our setting where both models are pre-trained and fixed. Forward Compatible Training (FCT) Ramanujan et al. (2022), while employing a converter similar to ours, relies on unavailable "side information". Jaeckle et al. (2023) removes this requirement but focuses on online backfilling with different data requirements and objectives. Crucially, all these works primarily target the image domain, whereas our method demonstrates broader applicability.

## 3 Methods: Embedding-Converter

This section introduces the proposed Embedding-Converter framework, designed to efficiently 'translate' embeddings from one model to another. While we demonstrate it using text embedding models, the framework is versatile and can handle various data types, including images and even multimodal data, for conversion of embedding models for them. Importantly, Embedding-Converter works with any embedding model, even those accessible only as prediction APIs with hidden internal details. This greatly expands its applicability, as many embedding models are provided solely via prediction-only APIs.

### 3.1 PROBLEM FORMULATION

The focus is to learn the transformation between two high dimensional spaces – specifically, we aim to develop a method for converting text embeddings generated by a source embedding model, denoted as $f$, into embeddings consistent with a target embedding model, denoted as $g$. Given a text passage $t \in \mathcal{T}$, where $\mathcal{T}$ represents the set of all text passages, we seek to construct a converter function $h$ such that $h(f(t)) \simeq g(t)$. This function maps embeddings from the source space $\mathbb{R}^{d_f}$ to the target space $\mathbb{R}^{d_g}$, where $d_f$ and $d_g$ represent the dimensions of the respective embedding spaces.

Our approach leverages unlabeled text data, denoted as $\mathcal{D} = \{t_1, t_2, ..., t_N\}$, comprising diverse text passages. Notably, this method does not require labeled data depicting inter-passage relationships. Any text corpus, such as the notable public ones like MSMarco (Bajaj et al., 2016) or Wikipedia corpus, can be utilized. The learning objective is to identify the optimal converter $h$ that maximizes the similarity between the converted embedding and the corresponding target embedding for any given text where the similarity measure can be defined using various criteria, including cosine similarity.

The proposed converter $h$ is designed as a unified model capable of handling any text $t \in \mathcal{T}$, irrespective of the dimensionality differences between the source and target embedding spaces. Consequently, distinct converter functions would be required for different source-target embedding model combinations. This contribution enables flexible utilization of various embedding models by facilitating seamless transitions between their respective embedding spaces.

### 3.2 LOSS FUNCTIONS

A straightforward approach for maximizing the similarity between converted and target embeddings is to employ a regression loss function, which minimizes the distance between the two embedding vectors. This can be expressed as:

$$\mathcal{L}_{reg} = \sum_{t=1}^{N} ||h(f(t)) - g(t)||_1. \tag{1}$$

While this equation utilizes the mean absolute error, alternative regression losses, such as mean squared error, could be employed as well. However, relying solely on regression loss is insufficient for accurate embedding conversion, as demonstrated in the ablation study (see Table 6). To enhance the fidelity of the conversion process, we introduce two supplementary loss functions designed to preserve both global and local relationships within the embedding spaces. The first, a global similarity loss (similar with (Park et al., 2019)), aims to maintain the overall distance between embeddings:

$$\mathcal{L}_{global} = \sum_{t_1, t_2 \in \mathcal{D}} |\text{Dist}(h(f(t_1)), h(f(t_2))) - \text{Dist}(g(t_1), g(t_2))|. \tag{2}$$

This loss function evaluates the difference in distances between pairs of randomly selected texts in both the converted and target embedding spaces, thereby encouraging the preservation of global structure (we utilize 1-cosine similarity as our distance metrics). The second, a local similarity loss, focuses on preserving neighborhood relationships:

$$\mathcal{L}_{local} = \sum_{t_1 \in \mathcal{D}} \sum_{t_2 \in NN_k(t_1)} |\text{Dist}(h(f(t_1)), h(f(t_2))) - \text{Dist}(g(t_1), g(t_2))|. \tag{3}$$

For each text $t_1$, this loss considers its $k$ nearest neighbors ($NN_k(t_1)$) (based on target embedding similarities) and penalizes discrepancies in their relative distances within the converted and target embedding spaces, thus promoting local neighborhood preservation ($k$ is set to 100 in experiments). The impact of these additional loss functions on the embedding conversion process is empirically evaluated in our experiments (see Table 6). Ultimately, the Embedding-Converter is jointly optimized using a weighted combination of these three loss functions:

$$h^* = \arg\min_h \mathcal{L}_{reg} + \alpha\mathcal{L}_{global} + \beta\mathcal{L}_{local}, \tag{4}$$

where $\alpha, \beta \geq 0$ are the hyperparameters controlling the relative importance of each loss component, which can be tuned using a validation set. Note that we employ batch training for all three loss functions to ensure computational efficiency.

### 3.3 IMPLEMENTATION DETAILS

The proposed Embedding-Converter can be implemented using any architecture capable of mapping $d_f$-dimensional vectors to $d_g$-dimensional vectors. In our experiments, we primarily employ a 4-layer perceptron with SELU activations (Klambauer et al., 2017). As discussed in Sec. 5, a Transformer architecture (Vaswani, 2017) yields slightly worse performance. Model selection and hyperparameter optimization are guided by the retrieval performance. That is, the model and hyperparameter configuration that maximizes retrieval effectiveness on a held-out validation set is selected. This criterion aligns with the practical objective of employing the converted embeddings in retrieval tasks. Hyper-parameters and additional training details can be found in Appendix. C.

## 4 EXPERIMENTS

This section presents empirical evaluations of the Embedding-Converter's performance across various scenarios. We first demonstrate the effectiveness in converting embeddings between different versions of the same model. Subsequently, we assess the ability to bridge the embedding spaces of distinct models. While our primary focus lies in evaluating the Embedding-Converter's impact on retrieval tasks, we also provide results on other embedding-dependent tasks, including text classification and semantic text similarity (STS) (Yang et al., 2018), to showcase broader applicability. A detailed comparison of the computational time and cost associated with traditional corpus re-embedding versus our proposed Embedding-Converter approach is presented in Appendix B.

### 4.1 EXPERIMENTAL SETTINGS

The Embedding-Converter is trained on a diverse set of text passages and queries drawn from 14 datasets in the BEIR benchmark (Thakur et al., 2021). We utilize a subset of the corpus data for training: half of the corpus for datasets with fewer than 1 million passages, and 500,000 randomly sampled passages for larger datasets (e.g., 10% of Fever, Climate-fever, and HotPotQA). To ensure adequate representation of query-side distributions, we include the entire query set from the MS-Marco dataset in the training data ($\sim$ 500K queries). Consequently, MSMarco is excluded from the in-domain evaluation to avoid potential bias. We evaluate the effectiveness of the Embedding-Converter in two distinct settings: in-domain and out-of-domain. In-domain evaluation assesses performance on the remaining 13 BEIR datasets using normalized Discounted Cumulative Gain at rank 10 (nDCG@10) as the retrieval metric (Järvelin & Kekäläinen, 2002). Out-of-domain generalization is evaluated on 12 datasets from the CQADupStack benchmark (Hoogeveen et al., 2015), which are entirely separate from the training data, using the same nDCG@10 metric. To further investigate the versatility of the Embedding-Converter, we extend our analysis beyond retrieval encompassing other embedding-dependent tasks, including text classification and STS. This evaluation provides insights into the generalizability and transferability of the converted embeddings across diverse applications. Dataset-specific details can be found in the Appendix D.

### 4.2 CONVERSION BETWEEN DIFFERENT MODEL VERSIONS

To evaluate the effectiveness of our Embedding-Converter in adapting to model updates, we utilize different versions of Google's Gecko text embedding models: gecko003 and gecko004 [1]. We begin by generating embeddings for the training dataset using both gecko003 and gecko004 models. This data is then used to train the Embedding-Converter, specifically to map embeddings from the gecko003 space to the gecko004 space. For evaluation, we apply the trained converter to transform the entire corpus of 13 BEIR datasets. We then assess retrieval performance in nDCG@10, comparing three different embedding sets: (1) the original gecko003 embeddings, (2) the original gecko004 embeddings, and (3) the gecko003 embeddings converted to the gecko004 space using Embedding-Converter. Crucially, we only convert the corpus embeddings; queries are consistently encoded using the target model (gecko004) across all conditions. This design choice allows us to isolate and specifically assess the impact of corpus embedding conversion on retrieval effectiveness, eliminating any confounding effects from query embedding variations. For source/target model evaluation, we use the source/target model for both query and corpus embedding, respectively.

---

[1] https://cloud.google.com/vertex-ai/generative-ai/docs/embeddings/get-text-embeddings

| Dataset | gecko003 → gecko004 | | | openai-3-small → gecko004 | | |
|---|---|---|---|---|---|---|
| | gecko003 (source) | gecko004 (target) | Embedding -Converter | openai-3-small (source) | gecko004 (target) | Embedding -Converter |
| Arguana | 0.5189 | 0.6070 | **0.6103** | 0.5530 | 0.6070 | **0.6049** |
| Climate-fever | 0.2540 | 0.3369 | **0.2959** | 0.2792 | 0.3369 | 0.2716 |
| DBPedia | 0.4128 | 0.4677 | **0.4322** | 0.4154 | 0.4677 | 0.4099 |
| Fever | 0.7431 | 0.8106 | **0.7786** | 0.7227 | 0.8106 | **0.7659** |
| FiQA | 0.4582 | 0.5481 | **0.5040** | 0.4048 | 0.5481 | **0.4393** |
| HotpotQA | 0.6248 | 0.6892 | 0.5923 | 0.6121 | 0.6892 | **0.6341** |
| NFCorpus | 0.3284 | 0.3503 | **0.3435** | 0.3314 | 0.3503 | **0.3479** |
| NQ | 0.5166 | 0.6058 | **0.5755** | 0.5254 | 0.6058 | **0.5653** |
| Quora | 0.8626 | 0.8621 | 0.8392 | 0.8881 | 0.8621 | 0.8346 |
| SciDocs | 0.1836 | 0.2041 | **0.1908** | 0.2092 | 0.2041 | 0.1995 |
| SciFact | 0.7221 | 0.7693 | **0.7601** | 0.7292 | 0.7693 | **0.7668** |
| Trec-covid | 0.7454 | 0.7840 | **0.8079** | 0.8285 | 0.7840 | **0.7983** |
| Touche | 0.2161 | 0.2565 | **0.2397** | 0.2723 | 0.2565 | **0.2706** |
| Average | 0.5067 | 0.5609 | **0.5362** | 0.5209 | 0.5609 | **0.5314** |

Table 1: In-domain retrieval performance (in nDCG@10) of the Embedding-Converter on 13 BEIR datasets. Two conversion scenarios are presented: (i) intra-model conversion between different versions of Google's Gecko model (gecko003 to gecko004), and (ii) inter-model conversion from OpenAI's text-embedding-3-small model to Google's gecko004. **Bold** represents better performance than the source or target models.

Table 1 demonstrates the effectiveness of the proposed Embedding-Converter in adapting to model updates. From gecko003 to gecko004, it yields a significant performance improvement over using the original gecko003 embeddings. Notably, the average performance of the converted embeddings is in the middle of source and target model performances for most datasets, while for some (e.g. Arguana, NFCorpus, Trec-Covid and SciFact) the Embedding-Converter performance is almost similar to the target model. This result highlights the capability of Embedding-Converter to efficiently transfer an entire corpus to a new embedding space with marginal performance degradation. Consequently, leveraging newer model versions becomes feasible without incurring the computational cost of re-embedding the entire corpus. As demonstrated in Appendix B, in practical scenarios, Embedding-Converter yields O(100) times cost and runtime savings. It constitutes significant implications to bring new paradigms for maintaining and updating large-scale retrieval systems.

### 4.3 Conversion across Different Model Families

To further showcase the versatility of the proposed Embedding-Converter, we extend our evaluation to scenarios involving conversions between different embedding models. Specifically, we investigate converting embeddings from OpenAI's text-embedding-3-small (openai-3-small) [2] to Google's gecko004. This experiment is particularly noteworthy as it involves models with different embedding dimensions – openai-3-small produces 1536-dimensional embeddings, while gecko004 produces 768-dimensional embeddings. Maintaining the same experimental setup as before, we evaluate the performance of a single Embedding-Converter trained to convert all corpora on the 13 BEIR datasets.

Table 1 demonstrates that even with inter-model conversion and a reduction in dimensionality, the Embedding-Converter still achieves significant mitigation of retrieval performance degradation. This result has important practical implications for machine learning developers. It enables efficient evaluation of new embedding models on existing corpora without the need for computationally expensive re-embedding. More specifically, Table 1(right) reveals that the target model outperforms the source model on 9 datasets, while the source model performs better on the remaining 4 datasets. Traditionally, determining which model is superior for a given dataset would require computing embeddings using both models. However, the Embedding-Converter offers an alternative approach. By comparing the performance with the source model, we can effectively approximate the comparisons

---

[2] https://platform.openai.com/docs/guides/embeddings

between the source and target models without incurring the computational cost of generating target embeddings. Our results demonstrate the effectiveness of this approach – the relative performance of the source and target models is accurately predicted by the Embedding-Converter on 11 out of the 13 datasets. This capability further highlights the value of the proposed Embedding-Converter. By facilitating seamless transitions between different embedding spaces, it promotes flexibility and reduces computational overhead in the development and deployment of embedding-based systems, while also offering a valuable tool for preliminary model comparison and selection.

## 4.4 GENERALIZATION TO OUT-OF-DOMAIN DATA

While the strong in-domain performance across 13 diverse datasets with a single Embedding-Converter is encouraging, evaluating its generalization capability on unseen data is paramount for practical applications. For generalizability to unseen tasks, out-of-domain performance is critical, as their specific data are likely to differ substantially from the datasets used in training. To assess the effectiveness in such scenarios, we evaluate its performance on 12 out-of-domain datasets from the CQADupStack benchmark, which are excluded from the training process.

| Dataset | gecko003 → gecko004 | | | openai-3-small → gecko004 | | |
|---|---|---|---|---|---|---|
| | gecko003 (source) | gecko004 (target) | Embedding -Converter | openai-3-small (source) | gecko004 (target) | Embedding -Converter |
| Android | 0.5258 | 0.5780 | **0.5687** | 0.5414 | 0.5780 | **0.5576** |
| English | 0.5019 | 0.5411 | **0.5163** | 0.5006 | 0.5411 | **0.5017** |
| Gaming | 0.6288 | 0.6720 | **0.6422** | 0.6125 | 0.6720 | **0.6287** |
| Gis | 0.3982 | 0.4503 | **0.4223** | 0.4055 | 0.4503 | **0.4178** |
| Mathematica | 0.2908 | 0.3621 | **0.3329** | 0.3053 | 0.3621 | **0.3265** |
| Physics | 0.4738 | 0.5291 | **0.4981** | 0.4615 | 0.5291 | **0.4832** |
| Programmers | 0.4455 | 0.5027 | **0.4766** | 0.4342 | 0.5027 | **0.4627** |
| Stats | 0.3531 | 0.4036 | **0.3715** | 0.3581 | 0.4036 | **0.3644** |
| Tex | 0.2958 | 0.3517 | **0.3201** | 0.2925 | 0.3517 | **0.3018** |
| Unix | 0.4362 | 0.4980 | **0.4622** | 0.4349 | 0.4980 | **0.4498** |
| Webmasters | 0.4297 | 0.4954 | **0.4698** | 0.4105 | 0.4954 | **0.4466** |
| Wordpress | 0.3453 | 0.3923 | **0.3701** | 0.3434 | 0.3923 | **0.3493** |
| Average | 0.4271 | 0.4814 | **0.4542** | 0.4250 | 0.4814 | **0.4408** |

Table 2: Out-of-domain retrieval performance (nDCG@10) of the Embedding-Converter on 12 CQADupStack datasets. Two conversion scenarios are presented: (i) intra-model conversion between different versions of Google's Gecko model (gecko003 to gecko004), and (ii) inter-model conversion from OpenAI's text-embedding-3-small model to Google's gecko004. **Bold** represents better performance than the source or target models.

Table 2 presents the results on this out-of-domain evaluation setting. Even in these challenging conditions, the Embedding-Converter consistently outperforms the source model, both within the same model family (gecko003 to gecko004) and across different models (openai-3-small to gecko004). Although the performance gap compared to the target model is larger than the gap in the in-domain setting, the Embedding-Converter still provides valuable means of estimating potential performance gains before committing to the computationally expensive process of re-embedding the entire corpus with the new model. It offers a preliminary performance guarantee when migrating to a new embedding model, enabling informed decision-making and resource allocation. Here, the relative performance of the source and target models is perfectly predicted by the Embedding-Converter.

## 4.5 PERFORMANCE ON OTHER TASKS BEYOND RETRIEVAL

While our primary focus has been on retrieval tasks, text embeddings are utilized in a wide range of applications. The MTEB benchmark (Muennighoff et al., 2022) encompasses diverse tasks such as classification, clustering, reranking, and STS, highlighting the versatility of embeddings. To assess the broader applicability of our Embedding-Converter, we evaluate its performance on two additional tasks: text classification and semantic text similarity. For classification, we use Toxic Conversation (cjadams, 2019) and Tweet Sentiment Extraction (Maggie, 2020) datasets. For se-

mantic text similarity, we use STS-13 (Agirre et al., 2013), STS-14 (Bandhakavi et al., 2014) and STS-22 (Chen et al., 2022) datasets.

| Task | Dataset | gecko003 → gecko004 | | | openai-3-small → gecko004 | | |
|------|---------|---------------------|----------|-------------------|---------------------------|----------|-------------------|
| | | gecko003 (source) | gecko004 (target) | Embedding -Converter | openai-3-small (source) | gecko004 (target) | Embedding -Converter |
| Classi-fication | Toxic Tweet | 0.9341 0.7261 | 0.9446 0.7535 | **0.9392** **0.7425** | 0.9380 0.7476 | 0.9446 0.7535 | **0.9410** 0.7434 |
| | Average | 0.8301 | 0.8491 | **0.8409** | 0.8428 | 0.8491 | 0.8422 |
| STS | STS-13 STS-14 STS-22 | 0.7712 0.7119 0.7019 | 0.8047 0.7403 0.7246 | **0.7982** **0.7359** **0.7080** | 0.8425 0.8001 0.6716 | 0.8047 0.7403 0.7246 | **0.8317** **0.7586** **0.6863** |
| | Average | 0.7283 | 0.7565 | **0.7474** | 0.7714 | 0.7565 | **0.7589** |

Table 3: Classification and STS performances of Embedding-Converter in two different settings: (i) within same model lineup but different versions (gecko003 → gecko004), (ii) across different model lineup (openai-3-small → gecko004) on 5 datasets. **Bold** represents better performance than the source or target models.

Table 3 presents the performance of the Embedding-Converter on classification and STS tasks. In the scenario of conversion from gecko003 to gecko004, the target model (gecko004) consistently outperforms the source model (gecko003), and the Embedding-Converter achieves performance levels between the two. This result demonstrates the converter's ability to effectively transfer relevant embedding properties for these tasks. For the openai-3-small to gecko004 conversion, the target model performs better in 3 out of 5 cases, while the source model is superior in the remaining 2 cases. Notably, the Embedding-Converter accurately predicts the relative performance of the source and target models in 4 out of these 5 cases. This further highlights the utility of the converter as a tool for preliminary model comparison, even across different model families. Overall, these results suggest that the converted embeddings successfully capture the semantic information encoded by the target model, enabling their effective utilization in diverse downstream tasks beyond retrieval. This generalization capability underscores the broader potential to facilitate efficient and flexible deployment of embedding models across a wide range of applications including unseen scenarios.

## 4.6 LEVERAGING FOR LATENCY REDUCTION

Thus far, we've focused on using the Embedding-Converter to transform corpus embeddings, a particularly valuable application when dealing with large corpora. The Embedding-Converter also offers significant advantages in scenarios where query latency is a critical concern. Often, deploying large embedding models for online query processing is impractical due to their high latency. While corpus embeddings can be pre-computed offline to mitigate latency concerns, query embeddings must be generated in real-time, making latency a significant bottleneck. Consequently, developers might resort to using smaller embedding models for both queries and corpora, even though larger models would yield better retrieval performance for the corpus.

The proposed Embedding-Converter offers a solution to this challenge. By decoupling corpus and query embedding models, we can leverage the superior performance of larger models for corpus embedding extraction while maintaining low query latency. This is achieved by employing a smaller model for initial query embedding generation and then utilizing the Embedding-Converter to map these embeddings to the space of the larger corpus embedding model.

To demonstrate this use case, we evaluate the performance of the Embedding-Converter when applied to queries instead of the corpus. The results, presented in Table 4, show that query conversion achieves comparable performance to corpus conversion in most cases (except the in-domain conversion case from openai-3-small). This finding highlights the potential of query conversion to improve retrieval performance in latency-constrained environments. By enabling the use of larger models for corpus embedding without sacrificing query speed, the Embedding-Converter offers a valuable tool for optimizing the trade-off between accuracy and efficiency in real-world retrieval systems.

| Methods | gecko003 → gecko004 | | openai-3-small → gecko004 | |
|---|---|---|---|---|
| | In-domain | Out-domain | In-domain | Out-domain |
| Source embedding model | 0.5067 | 0.4271 | 0.5209 | 0.4250 |
| Target embedding model | 0.5609 | 0.4814 | 0.5609 | 0.4814 |
| Corpus converter | **0.5362** | **0.4542** | **0.5314** | **0.4408** |
| Query converter | **0.5263** | **0.4348** | 0.5171 | **0.4342** |

Table 4: Embedding-Converter on query converting scenarios with two different settings: (i) within same model lineup but different versions (gecko003 → gecko004), (ii) across different model lineup (openai-3-small → gecko004). **Bold** represents better performance than the source or target models.

## 5 DISCUSSIONS

### 5.1 FURTHER ANALYSES OF EMBEDDING-CONVERTER PERFORMANCE

While evaluating the Embedding-Converter on downstream tasks, as presented in Section 4, provides valuable insights into its practical utility, a comprehensive assessment necessitates further analysis of its ability to accurately align embedding spaces. This section delves into this aspect by employing quantitative metrics, specifically distance-based measures, to evaluate the converter's performance independent of specific downstream tasks.

We conduct this analysis by examining both global and local distances among corpus embeddings. Global distances provide a macroscopic view of the embedding space, capturing its overall structure and organization. Conversely, local distances offer a microscopic perspective, focusing on the preservation of relationships within local neighborhoods of the embedding space. By analyzing both the global and local distances, we gain a comprehensive understanding of the Embedding-Converter's efficacy in accurately mapping embeddings between different models while preserving the inherent structure of the embedding spaces.

| Settings | Methods | gecko003 → gecko004 | | openai-3-small → gecko004 | |
|---|---|---|---|---|---|
| | | Global distance | Local distance | Global distance | Local distance |
| In-domain | Source model | 0.1053 | 0.0246 | 0.2346 | 0.1260 |
| | Converter | **0.0393** | **0.0163** | **0.0191** | **0.0205** |
| Out-domain | Source model | 0.0805 | 0.0217 | 0.1811 | 0.1291 |
| | Converter | **0.0325** | **0.0176** | **0.0179** | **0.0195** |

Table 5: Comparison of global and local distance metrics (i.e., Eq. 2 and 3, lower the better) for the Embedding-Converter on 13 BEIR and 12 CQADupStack datasets. Two conversion scenarios are presented: (i) intra-model conversion between different versions of Google's Gecko model (gecko003 to gecko004), and (ii) inter-model conversion from OpenAI's text-embedding-3-small model to Google's gecko004. **Bold** represents better performance than the source model.

Table 5 shows that the Embedding-Converter effectively aligns both global and local distances between the converted embeddings and the target embeddings, preserving meaningful positioning. This result underscores the ability to accurately capture and replicate the inherent structural properties of the target embedding space, further validating efficacy in facilitating cross-model mapping. Further experiments evaluating our method in diverse practical settings, including reverse conversion, handling mixed embeddings, and bridging open-source to black-box models, can be found in the appendix.

### 5.2 ABLATION STUDIES

We investigate the contributions of different loss functions and architectural choices in the overall performance. Recall that the Embedding-Converter is trained using a combination of three loss functions: a regression loss ($\mathcal{L}_{reg}$), a global loss ($\mathcal{L}_{global}$), and a local loss ($\mathcal{L}_{local}$). In our ablation studies, we analyze the effect of removing these loss components. Additionally, we explore

the impact of architectural variations by replacing the default multi-layer perceptron (MLP) with a Transformer architecture. By systematically analyzing the effects of these modifications, we aim to identify the key components driving the Embedding-Converter's performance and gain insights into their individual contributions.

| Variants | Performances | | |
|---|---|---|---|
| | Global distance | Local distance | Retrieval |
| No $\mathcal{L}_{global}$ & $\mathcal{L}_{local}$ | 0.0452 | 0.0237 | 0.5219 |
| Transformer architecture | 0.0233 | 0.0211 | 0.5273 |
| Small networks (20% parameters) | 0.0203 | 0.0192 | 0.5263 |
| Larger networks (5x parameters) | **0.0177** | 0.0164 | 0.5329 |
| Only with MSMarco | 0.0351 | 0.0227 | 0.5194 |
| No variants | **0.0177** | **0.0163** | **0.5369** |

Table 6: Ablation studies across different variants of Embedding-Converter. Global distance, local distance, and Retrieval performances are evaluated on out-domain retrieval tasks (with 12 CQADup-Stack datasets). Here, we use the Embedding-Converter from gecko003 to gecko004.

Table 6 summarizes the results of our ablation studies, highlighting key factors influencing the Embedding-Converter's performance:

- **Loss functions**: Both global and local loss functions are crucial. Removing them leads to performance degradation, especially in distance metrics, underscoring their complementary roles.

- **Architecture variations**: The choice of Transformer vs. MLP impacts performance, suggesting sensitivity to architectural design choices, given sufficient model capacity and proper training.

- **Model size**: Smaller models (compared with original Embedding-Converter) exhibit slightly lower performance due to reduced capacity for capturing complex relationships in embedding spaces. Larger models perform consistent with original Embedding-Converter.

- **Data diversity**: Diverse training data significantly improves performance by enhancing generalization and coverage across the embedding space (Fig. 2(a)). Relying solely on MSMarco is insufficient for broad coverage.

## 6 CONCLUSIONS

This paper addresses the critical challenge of achieving seamless compatibility between different embedding models. The lack of such compatibility hinders both machine learning practitioners, who face difficulties in navigating model updates and selecting optimal models, and the overall robustness of deployed systems. To overcome this limitation, we propose a unified Embedding-Converter, capable of efficiently translating between different embedding models. In its design, we address the unique challenges of learning how to convert embeddings efficiently via judicious chosen training mechanisms and show that in many scenarios, the end-to-end performance with converted embeddings can be largely preserved. This contribution can empower practitioners with the flexibility to effortlessly transition between models, fostering greater experimentation and facilitating the adoption of improved model versions. Furthermore, Embedding-Converter can encourage a paradigm shift in model development practices. By emphasizing the importance of providing converters alongside new model releases, a more user-centric approach is enabled for seamless migration from previous versions. This fosters a dynamic and evolving ecosystem for embedding models, where innovation and user experience are prioritized. This, in turn, contributes to a more robust and user-friendly environment for developing and deploying embedding-based applications.

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

APPENDIX

## A    VISUALIZATION OF EMBEDDING SPACES

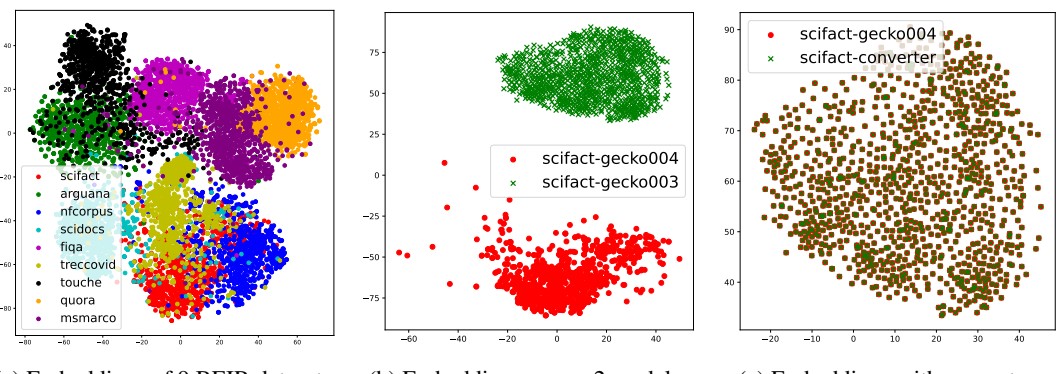

(a) Embeddings of 9 BEIR datasets    (b) Embeddings across 2 models    (c) Embeddings with converter

Figure 2: t-SNE visualization of embedding spaces across different corpora and models. (a) Embeddings of 9 diverse corpora from the BEIR datasets, highlighting the variability in embedding distributions across different datasets. (b) Comparison of gecko003 and gecko004 embeddings for the SciFact dataset, showcasing how different the embedding spaces between different model versions can be for the same dataset. (c) Embeddings of the gecko004 model and embeddings converted from gecko003 using the Embedding-Converter. The high degree of overlap indicates the successful alignment of embedding spaces achieved by the Embedding-Converter.

## B    COMPUTATIONAL COMPLEXITY

To quantify the computational benefits of the proposed Embedding-Converter, we analyze both the computational cost and processing time across various scenarios. We assume a consistent document length of 256 tokens and utilize three different embedding models with varying pricing and request-per-minute (RPM) limitations, as detailed below. This analysis provides a concrete assessment of the efficiency gains achieved by leveraging the Embedding-Converter compared to the traditional approach of re-embedding the entire corpus.

- Openai-3-large:
    - Price: $0.065 / 1M tokens [3]
    - RPM: 1M tokens [4] (with free tier)
- Openai-3-small:
    - Price: $0.010 / 1M tokens
    - RPM: 1M tokens (with free tier)
- Gecko004:
    - Price: $0.00002 / 1K characters (i.e., $0.08 / 1M tokens when we assume average 4 characters per one token)[5]
    - RPM: 7500 inputs [6]

While OpenAI's embedding API can offer higher RPM at its highest tier, potentially reducing computation time, it still remains significantly slower than our Embedding-Converter. Furthermore, the

---

[3]https://openai.com/api/pricing/
[4]https://platform.openai.com/docs/guides/rate-limits/usage-tiers?context=tier-free
[5]https://cloud.google.com/vertex-ai/generative-ai/pricing
[6]https://cloud.google.com/vertex-ai/generative-ai/docs/quotas#text-embedding-limits

| Embedding model | Corpus size | Computational cost | | Computational time | |
|---|---|---|---|---|---|
| | | Baseline | Embedding-Converter | Baseline | Embedding-Converter |
| Openai-3-large | 1B | $16640 | $185 | 4266 hours | 37 hours |
| | 50M | $832 | $10 | 213 hours | 1.9 hours |
| Openai-3-small | 1B | $2560 | $115 | 4266 hours | 23 hours |
| | 50M | $128 | $6 | 213 hours | 1.2 hours |
| gecko004 | 1B | $20480 | $75 | 2222 hours | 15 hours |
| | 50M | $1024 | $4 | 111 hours | 0.8 hours |

Table 7: Computational cost and time comparisons with Embedding-Converter for different corpus sizes. Baseline refers to recomputation of the entire corpus using the correspond embedding models.

cost per API call is consistent across all tiers, offering no cost advantage for higher RPM usage. In contrast, our Embedding-Converter exhibits remarkable efficiency gains with modest compute, and even without low-level engineering optimizations. It can process a corpus of size 50 million in under 2 hours, including data loading. For inference alone, using a pre-trained Embedding-Converter takes a mere 20 minutes to process the same corpus with openai-3-small. This represents a speed improvement of over 100x compared to traditional corpus re-embedding.

The Embedding-Converter utilizes 2 V100 GPUs, incurring an hourly cost of $4.96 on Cloud[7]. This results in a computational cost reduction exceeding 100x compared to directly generating embeddings with the target model. These findings underscore the substantial efficiency gains offered by our Embedding-Converter. It provides a compelling solution for migrating to new embedding models, enabling both cost and time savings, especially when handling large-scale corpora.

## C  HYPER-PARAMETERS & TRAINING DETAILS

As the Embedding-Converter architecture, we employ a 4-layer multi-layer perceptron (MLP) model with hidden state dimensions of (5x output-dimension, 5x output-dimension, 5x output-dimension, output-dimension). Therefore, when converting between gecko003 and gecko004 embeddings, the Embedding-converter comprises 35 million parameters. We utilize the SELU activation function and apply L2 normalization to the output. Optimization is performed using the Adam optimizer with a learning rate of 0.001. Training proceeds for 50,000 iterations, with batches sampled uniformly from each of the 14 BEIR datasets to mitigate dataset bias and enhance coverage. This balanced sampling strategy involves selecting an equal number of batches from each dataset, with a batch size of 64. Validation performance is evaluated every 250 iterations, and the model yielding the best validation performance is selected. In all experiments, we utilized the Scifact dataset (comprising 1109 queries, 1258 labels, and 5183 corpus passages) for validation data. The hyperparameters $\alpha$ and $\beta$, which control the weighting of the global and local loss functions, are tuned within the range of $[0.01, 1.0]$. In most cases, $\alpha = \beta = 0.1$ consistently yielded strong results. For the local distance loss, we consistently set the neighborhood size $(k)$ to 100 across all experiments.

## D  DATA STATISTICS

### D.1  BEIR DATASETS

---

[7]https://cloud.google.com/compute/gpus-pricing

| Datasets | Number of queries | Number of test pairs | Number of corpus |
|---|---|---|---|
| Arguana | 1406 | 1406 | 8674 |
| Climate-fever | 1535 | 4681 | 5416593 |
| DBPedia | 467 | 49188 | 4635922 |
| Fever | 123142 | 148022 | 5416568 |
| FiQA | 6648 | 15872 | 57638 |
| HotPotQA | 97852 | 184810 | 5233329 |
| NFCorpus | 3237 | 122909 | 3633 |
| NQ | 3452 | 4201 | 2681468 |
| Quora | 15000 | 23301 | 522931 |
| SciDocs | 1000 | 29928 | 25657 |
| SciFact | 1109 | 1258 | 5183 |
| Trec-Covid | 50 | 66336 | 171332 |
| Touche | 49 | 2214 | 382545 |

Table 8: The statistics of 13 BEIR datasets (sorted by the alphabetical order).

## D.2 CQADUPSTACK DATASETS

| Datasets | Number of queries | Number of test pairs | Number of corpus |
|---|---|---|---|
| Android | 699 | 1696 | 22998 |
| English | 1570 | 3765 | 40221 |
| Gaming | 1595 | 2263 | 45301 |
| Gis | 885 | 1114 | 37637 |
| Mathematica | 804 | 1358 | 16705 |
| Physics | 1039 | 1933 | 38316 |
| Programmers | 876 | 1675 | 32176 |
| Stats | 652 | 913 | 42269 |
| Tex | 2906 | 5154 | 68184 |
| Unix | 1072 | 1693 | 47382 |
| Webmasters | 506 | 1395 | 17405 |
| Wordpress | 541 | 744 | 48605 |

Table 9: The statistics of 12 CQADupStack datasets (sorted by alphabetical order).

## D.3 STS AND CLASSIFICATION DATASETS

| Tasks | Datasets | Number of train samples | Number of test samples | Number of classes |
|---|---|---|---|---|
| Classification | Toxic | 50000 | 50000 | 2 |
| | Tweet | 27481 | 3534 | 3 |
| STS | STS-13 | - | 1500 | - |
| | STS-14 | - | 3750 | - |
| | STS-22 | - | 197 | - |

Table 10: The statistics of 2 classification and 3 STS datasets.

# E  ADDITIONAL EXPERIMENTS

## E.1  CONVERTING FROM NEW TO OLD EMBEDDING MODELS

To further validate the versatility of our Embedding-Converter, we conducted experiments where the conversion direction was reversed: from a newer embedding model to an older one. This scenario might arise when developers need to "downgrade" their embedding models due to resource constraints or compatibility requirements. Specifically, we used Gecko-004 as the source model and Gecko-003 as the target, effectively reversing the conversion direction presented in Table 1 (left) and Table 2 (left) of the main manuscript.

| Dataset | gecko004 → gecko003 | | | gecko004 → openai-3-small | | |
|---|---|---|---|---|---|---|
| | gecko004 (source) | gecko003 (target) | Embedding -Converter | gecko004 (source) | openai-3-small (target) | Embedding -Converter |
| Arguana | 0.6070 | 0.5189 | 0.5148 | 0.6070 | 0.5530 | 0.5713 |
| Climate-fever | 0.3369 | 0.2540 | 0.2905 | 0.3369 | 0.2792 | 0.2931 |
| DBPedia | 0.4677 | 0.4128 | 0.3979 | 0.4677 | 0.4154 | 0.3898 |
| Fever | 0.8106 | 0.7431 | 0.7327 | 0.8106 | 0.7227 | 0.6972 |
| FiQA | 0.5481 | 0.4582 | 0.4824 | 0.5481 | 0.4048 | 0.4507 |
| HotpotQA | 0.6892 | 0.6248 | 0.5794 | 0.6892 | 0.6121 | 0.5519 |
| NFCorpus | 0.3503 | 0.3284 | 0.3347 | 0.3503 | 0.3314 | 0.3318 |
| NQ | 0.6058 | 0.5166 | 0.5147 | 0.6058 | 0.5254 | 0.5151 |
| Quora | 0.8621 | 0.8626 | 0.8369 | 0.8621 | 0.8881 | 0.8396 |
| SciDocs | 0.2041 | 0.1836 | 0.1743 | 0.2041 | 0.2092 | 0.1928 |
| SciFact | 0.7693 | 0.7221 | 0.7227 | 0.7693 | 0.7292 | 0.7074 |
| Trec-covid | 0.7840 | 0.7454 | 0.7187 | 0.7840 | 0.8285 | 0.8278 |
| Touche | 0.2565 | 0.2161 | 0.2423 | 0.2565 | 0.2723 | 0.2684 |
| Average | 0.5609 | 0.5067 | 0.5032 | 0.5609 | 0.5209 | 0.5105 |

Table 11: In-domain retrieval performance (in nDCG@10) of the Embedding-Converter on 13 BEIR datasets. Two conversion scenarios are presented: (i) intra-model conversion between different versions of Google's Gecko model (gecko004 to gecko003), and (ii) inter-model conversion from Google's gecko004 to OpenAI's text-embedding-3-small model.

The results, shown in Table 11 (left) and Table 12 (left), demonstrate that the performance of our Embedding-Converter remains remarkably consistent with that of the target (older) model. This finding underscores the flexibility of our approach and its ability to support both upgrading and downgrading of embedding models, catering to a wider range of practical use cases.

## E.2 CONVERTING FROM SMALLER DIMENSIONAL EMBEDDING TO LARGER DIMENSIONAL EMBEDDINGS

While the main manuscript focused on embedding conversion between models with the same dimensionality or where the target model has smaller dimensionality, we further investigated the scenario where the target model possesses larger embedding dimensions. This represents another challenging yet practical use case, as newer embedding models often exhibit increased dimensionality.

To evaluate this scenario, we trained and evaluated our Embedding-Converter with gecko004 (768 dimensions) as the source model and openai-3-small (1536 dimensions) as the target.

| Dataset | gecko004 → gecko003 | | | gecko004 → openai-3-small | | |
|---|---|---|---|---|---|---|
| | gecko004 (source) | gecko003 (target) | Embedding -Converter | gecko004 (source) | openai-3-small (target) | Embedding -Converter |
| Android | 0.5780 | 0.5258 | 0.5172 | 0.5780 | 0.5414 | 0.5374 |
| English | 0.5411 | 0.5019 | 0.4785 | 0.5411 | 0.5006 | 0.4844 |
| Gaming | 0.6720 | 0.6288 | 0.6175 | 0.6720 | 0.6125 | 0.6052 |
| Gis | 0.4503 | 0.3982 | 0.4008 | 0.4503 | 0.4055 | 0.3951 |
| Mathematica | 0.3621 | 0.2908 | 0.2879 | 0.3621 | 0.3053 | 0.2984 |
| Physics | 0.5291 | 0.4738 | 0.4750 | 0.5291 | 0.4615 | 0.4670 |
| Programmers | 0.5027 | 0.4455 | 0.4479 | 0.5027 | 0.4342 | 0.4460 |
| Stats | 0.4036 | 0.3531 | 0.3444 | 0.4036 | 0.3581 | 0.3384 |
| Tex | 0.3517 | 0.2958 | 0.2849 | 0.3517 | 0.2925 | 0.2879 |
| Unix | 0.4980 | 0.4362 | 0.4287 | 0.4980 | 0.4349 | 0.4329 |
| Webmasters | 0.4954 | 0.4297 | 0.4345 | 0.4954 | 0.4105 | 0.4338 |
| Wordpress | 0.3923 | 0.3453 | 0.3289 | 0.3923 | 0.3434 | 0.3334 |
| Average | 0.4814 | 0.4271 | 0.4205 | 0.4814 | 0.4250 | 0.4217 |

Table 12: Out-of-domain retrieval performance (nDCG@10) of the Embedding-Converter on 12 CQADupStack datasets. Two conversion scenarios are presented: (i) intra-model conversion between different versions of Google's Gecko model (gecko004 to gecko003), and (ii) inter-model conversion from Google's gecko004 to OpenAI's text-embedding-3-small model.

The results, presented in Table 11 (right) and Table 12 (right), demonstrate that our method successfully handles this conversion with only marginal performance degradation. This finding further reinforces the robustness and generalizability of the Embedding-Converter, showcasing its ability to effectively bridge embedding spaces even when the target dimensionality exceeds that of the source.

### E.3 EMBEDDING-CONVERTER WITH MIXED EMBEDDINGS

Real-world applications often involve dynamic corpus sets where new documents are continuously added. Embedding-Converter also offers a significant advantage in such scenarios. Instead of requiring the conversion of new documents into the source embedding space before generating target embeddings, we can directly embed them using the target embedding model. This results in a corpus containing a mixture of converted embeddings (from older documents) and new embeddings (from recently added documents). To evaluate the effectiveness of our embedding converter in this mixed setting, we conducted experiments where half of the corpus embeddings were randomly replaced with target embeddings.

| Dataset | gecko003 | gecko004 | gecko003 → gecko004 | |
| | | | Embedding-Converter | |
| | (source) | (target) | Standard | Mixed |
|---|---|---|---|---|
| Arguana | 0.5189 | 0.6070 | 0.6103 | 0.6082 |
| Climate-fever | 0.2540 | 0.3369 | 0.2959 | 0.3124 |
| DBPedia | 0.4128 | 0.4677 | 0.4322 | 0.4486 |
| Fever | 0.7431 | 0.8106 | 0.7786 | 0.7946 |
| FiQA | 0.4582 | 0.5481 | 0.5040 | 0.5196 |
| HotpotQA | 0.6248 | 0.6892 | 0.5923 | 0.6410 |
| NFCorpus | 0.3284 | 0.3503 | 0.3435 | 0.3466 |
| NQ | 0.5166 | 0.6058 | 0.5755 | 0.5435 |
| Quora | 0.8626 | 0.8621 | 0.8392 | 0.8304 |
| SciDocs | 0.1836 | 0.2041 | 0.1908 | 0.1963 |
| SciFact | 0.7221 | 0.7693 | 0.7601 | 0.7671 |
| Trec-covid | 0.7454 | 0.7840 | 0.8079 | 0.7865 |
| Touche | 0.2161 | 0.2565 | 0.2397 | 0.2481 |
| Average | 0.5067 | 0.5609 | 0.5362 | 0.5419 |

Table 13: In-domain retrieval performance (in nDCG@10) of the Embedding-Converter on 13 BEIR datasets from gecko003 to gecko004. Two conversion scenarios are presented: (i) Standard: with 100% converted corpus, (ii) Mixed: with 50% converted corpus and 50% target corpus.

The results, presented in Table 13 and 14, demonstrate that performance in this mixed setting actually surpasses the scenario where all embeddings are converted. This observation highlights two key strengths of our approach: (i) Compatibility: The converted embeddings seamlessly integrate with the new embeddings, indicating strong compatibility between the two spaces. (ii) Generalizability: Our embedding converter effectively handles the practical scenario of mixed embeddings, further validating its robustness and applicability.

| Dataset | gecko003 → gecko004 | | | |
| --- | --- | --- | --- | --- |
| | gecko003 | gecko004 | Embedding-Converter | |
| | (source) | (target) | Standard | Mixed |
| Android | 0.5258 | 0.5780 | 0.5687 | 0.5632 |
| English | 0.5019 | 0.5411 | 0.5163 | 0.5255 |
| Gaming | 0.6288 | 0.6720 | 0.6422 | 0.6547 |
| Gis | 0.3982 | 0.4503 | 0.4223 | 0.4394 |
| Mathematica | 0.2908 | 0.3621 | 0.3329 | 0.3490 |
| Physics | 0.4738 | 0.5291 | 0.4981 | 0.5148 |
| Programmers | 0.4455 | 0.5027 | 0.4766 | 0.4877 |
| Stats | 0.3531 | 0.4036 | 0.3715 | 0.3846 |
| Tex | 0.2958 | 0.3517 | 0.3201 | 0.3323 |
| Unix | 0.4362 | 0.4980 | 0.4622 | 0.4775 |
| Webmasters | 0.4297 | 0.4954 | 0.4698 | 0.4781 |
| Wordpress | 0.3453 | 0.3923 | 0.3701 | 0.3807 |
| Average | 0.4271 | 0.4814 | 0.4542 | 0.4656 |

Table 14: Out-of-domain retrieval performance (nDCG@10) of the Embedding-Converter on 12 CQADupStack datasets from gecko003 to gecko004. Two conversion scenarios are presented: (i) Standard: with 100% converted corpus, (ii) Mixed: with 50% converted corpus and 50% target corpus.

### E.4 EMBEDDING-CONVERTER WITH MULTIPLE VERSIONS OF EMBEDDING MODELS

The landscape of embedding models is constantly evolving, with new versions frequently released. This raises the practical challenge of converting embeddings across multiple model iterations. For example, a user might need to transition from gecko003 to GTE-Large and then to gecko004.

While a direct conversion from gecko003 to gecko004 is possible, we also investigated the feasibility of utilizing a sequence of converters: gecko003 to GTE-Large followed by GTE-Large to gecko004. This approach could be advantageous in scenarios where direct conversion is computationally expensive or when intermediate embeddings are required.

| Dataset | gecko003 → gecko004 | | | |
| --- | --- | --- | --- | --- |
| | gecko003 | gecko004 | Embedding-Converter | |
| | (source) | (target) | Direct | Multiple |
| Arguana | 0.5189 | 0.6070 | 0.6103 | 0.5812 |
| FiQA | 0.4582 | 0.5481 | 0.5040 | 0.4903 |
| NFCorpus | 0.3284 | 0.3503 | 0.3435 | 0.3470 |
| Quora | 0.8626 | 0.8621 | 0.8392 | 0.8361 |
| SciDocs | 0.1836 | 0.2041 | 0.1908 | 0.1953 |
| SciFact | 0.7221 | 0.7693 | 0.7601 | 0.7626 |
| Trec-covid | 0.7454 | 0.7840 | 0.8079 | 0.7580 |
| Touche | 0.2161 | 0.2565 | 0.2397 | 0.2358 |
| Average | 0.5044 | 0.5609 | 0.5369 | 0.5258 |

Table 15: In-domain retrieval performance (in nDCG@10) of the Embedding-Converter on 8 BEIR datasets from gecko002 to gecko004. Two conversion scenarios are presented: (i) Direct: converting gecko003 to gecko004 directly, (ii) Multiple: converting gecko003 to GTE-Large first and then converting GTE-Large to gecko004.

Our experiments compared the performance of these two strategies in Table 15. While direct conversion yielded slightly better results, the difference was marginal. This observation highlights the flexibility of our Embedding-Converter and its ability to effectively handle multi-version embedding conversions, offering a practical solution for navigating the evolving landscape of embedding models.

## E.5 CONVERTING OPEN-SOURCE MODEL TO BLACK-BOX MODEL

To further demonstrate the versatility of our Embedding-Converter, we extended our evaluation to scenarios involving conversion between open-source and black-box embedding models. This is crucial for ensuring compatibility and facilitating transitions across different model ecosystems. Specifically, we converted embeddings from the open-source GTE-Large model Li et al. (2023) to Google's black-box gecko004 model.

| Dataset | GTE-Large → gecko004 | | |
|---|---|---|---|
| | GTE-Large (source) | gecko004 (target) | Embedding-Converter |
| Arguana | 0.5928 | 0.6070 | 0.6081 |
| FiQA | 0.4434 | 0.5481 | 0.5059 |
| NFCorpus | 0.3391 | 0.3503 | 0.3478 |
| Quora | 0.8824 | 0.8621 | 0.8391 |
| SciDocs | 0.2330 | 0.2041 | 0.2080 |
| SciFact | 0.7402 | 0.7693 | 0.7689 |
| Trec-covid | 0.7053 | 0.7840 | 0.7628 |
| Touche | 0.2237 | 0.2565 | 0.2431 |
| Average | 0.5200 | 0.5477 | 0.5355 |

Table 16: In-domain retrieval performance (in nDCG@10) of the Embedding-Converter on 8 BEIR datasets across inter-model conversion from GTE-Large to Google's gecko004 model.

As shown in Table 16, the Embedding-Converter successfully bridges these two models across various BEIR datasets, maintaining strong performance. This result underscores the generalizability of our approach and its ability to handle diverse conversion scenarios, including those involving both open-source and proprietary embedding models.

