# OpenReview forum: "Embedding-Converter: A Unified Framework for Cross-Model Embedding Transformation"
_ICLR.cc/2025/Conference — Submitted to ICLR 2025_

### Official Review · Reviewer_h1G5 · 2024-10-29

**Soundness:** 2
**Presentation:** 1
**Contribution:** 2
**Rating:** 3
**Confidence:** 4

**Summary:**

The paper addresses a practical industrial problem - how to quickly determine if a new embedding model is better when we have an existing pipeline built on a specific embedding model.
The paper proposes a straightforward approach - training a converter between embedding spaces and observing the effects in the pipeline.

**Strengths:**

The paper raises a valuable question with significant industrial relevance.

**Weaknesses:**

* The paper's approach is rather direct and simple, lacking deeper analysis of the problem. For instance, it doesn't explore whether mapping from higher to lower dimensional embeddings is equivalent to mapping from lower to higher dimensions.
* The paper fails to establish clear criteria for evaluating new embeddings - the performance after conversion may be inconsistent with direct usage of new embeddings, potentially being better or worse, and lacks a golden standard for assessment.
* The study lacks experiments on mapping from stronger embeddings to weaker ones.

**Questions:**

N/A

---

> ### Author Response · Authors · 2024-11-18
> **Response to Reviewer h1G5's comments**
>
> We appreciate your acknowledgment of the method's practical significance and your detailed review of our paper. Please find our response to your comments below.  A revised manuscript will be submitted soon. We are happy to address any further questions you may have.
>
> **Answer 1:** Following the reviewer's suggestion, we have expanded our experiments in the revised manuscript to include the conversion from gecko003 (768 dimensions) to openai-3-small (1536 dimensions). This new addition provides a direct comparison to the original results showcasing conversion from openai-3-small to gecko004, effectively demonstrating the embedding converter's efficacy in handling both lower-to-higher and higher-to-lower dimensional transformations. This further strengthens our claim regarding the versatility and robustness of the proposed method.
>
> **Answer 2:** To clarify our evaluation methodology, we emphasize that all tables directly compare the performance of our embedding converter against the "golden standard" of directly using the new (target) embeddings.
>
> For instance, in Table 1, the columns labeled "gecko004 (target)" represent the retrieval performance achieved when utilizing the new gecko004 embeddings without any conversion.  These results serve as the benchmark for assessing the effectiveness of our approach. The "Embedding-converter" column then presents the performance achieved when applying our method to convert embeddings to the gecko004 space.
>
> This direct comparison allows us to clearly demonstrate the ability of our embedding converter to achieve comparable performance to using the new embeddings directly, while avoiding the associated costs of full model migration.
>
> **Answer 3:** While the primary objective of our Embedding-converter is to facilitate efficient migration to newer models for improved performance, we acknowledge the reviewer's request to explore the reverse scenario. Therefore, in this rebuttal, we have expanded our experiments to include conversions from newer models to older models. This demonstrates the versatility of our approach and its ability to handle bidirectional conversions effectively, further strengthening the case for its general applicability.

---

> > ### Author Response · Authors · 2024-12-02
> > **Reminder to author-reviewer discussion**
> >
> > We appreciate your insightful comments and have carefully addressed each one in our rebuttal and revised manuscript. We hope our revisions have clarified the concerns you raised. Please take a look at our response and the updated paper, and let us know if you have any further questions.

---

### Official Review · Reviewer_MnsG · 2024-11-02

**Soundness:** 2
**Presentation:** 2
**Contribution:** 2
**Rating:** 5
**Confidence:** 1

**Summary:**

This paper introduces a unified framework called "Embedding-Converter" aimed at efficiently converting embeddings between different models, addressing the high cost associated with re-computing entire dataset embeddings when switching or upgrading models. The main contributions include:

Framework Design: The Embedding-Converter acts as a "translator for embedding spaces," enabling seamless conversion of embeddings between various models, thus eliminating the need to regenerate embeddings. This allows for quick and low-cost model switching or version upgrades and is applicable to different types of data across tasks and domains.

Performance Validation: Through experiments, Embedding-Converter achieves performance close to the target model on large datasets and, in some tasks, even surpasses the source model's performance. The method saves over 100 times in computational cost and time compared to traditional methods, highlighting its potential in model migration.

Multi-Task Extension: Embedding-Converter is not only suitable for retrieval tasks but also performs well in other tasks like text classification and semantic similarity, demonstrating its broad applicability.

Efficiency and Flexibility: This framework supports efficient vector transformation in high-dimensional spaces, preserving local and global structures among embeddings, enabling a smooth transition during model migration.

**Strengths:**

Reduction in Cost and Time for Model Migration: Traditional methods require re-embedding entire datasets when switching embedding models, which is costly and time-consuming, especially for large datasets. Embedding-Converter allows efficient conversion of embeddings between models, eliminating the need for re-embedding and reducing the cost and time of model migration by over 100 times. This efficiency is crucial for applications requiring frequent model updates.

Support for Cross-Model and Cross-Version Conversion: This framework can convert embeddings not only between different versions of the same model (e.g., Google’s Gecko003 to Gecko004) but also between entirely different models (e.g., from an OpenAI model to a Google model). This flexibility enables users to switch to the best-performing model for their task without re-computation.

High-Quality Embedding Conversion: Experiments show that Embedding-Converter generates converted embeddings that perform close to or even better than the source model on multiple downstream tasks, while also maintaining target model performance. By leveraging various loss functions, such as global and local similarity losses, the framework ensures that the converted embeddings structurally align well with the target model.

Broad Applicability Across Tasks: Besides retrieval tasks, the framework also excels in text classification and semantic similarity tasks, demonstrating its versatility. This suggests that Embedding-Converter can adapt to various data types and tasks, with the potential for cross-domain applications.

Reduced Latency and Improved Real-Time Performance: Embedding-Converter can convert query embeddings in low-latency scenarios, allowing users to employ larger models for dataset embeddings and smaller models for real-time queries, significantly improving both the speed and accuracy of real-time tasks.

**Weaknesses:**

I am not familiar with this direction.

**Questions:**

N/A

I will see the comments of other reviewers.

---

> ### Author Response · Authors · 2024-11-18
> **Response to Reviewer MnsG's comments**
>
> We appreciate the reviewer acknowledging the key strengths of our work, particularly: (i) The significant reduction in cost and time associated with model migration. (ii) The demonstrated generalizability of our embedding converter across different models and model versions. (iii) The compelling experimental results supporting the effectiveness of our approach. (iv) The broad applicability of our method to various real-world scenarios. We are open to any further questions or suggestions the reviewer may have and are committed to incorporating your valuable feedback into the revised manuscript.

---

> > ### Comment · Reviewer_MnsG · 2024-11-23
> > **To response**
> >
> > After reading other reviewers' comments, I maintain my score.

---

### Official Review · Reviewer_q2HY · 2024-11-02

**Soundness:** 3
**Presentation:** 3
**Contribution:** 1
**Rating:** 5
**Confidence:** 5

**Summary:**

The paper introduces the concept of an embedding converter: a model that transforms the embedding space between two embedding models. The motivation stems from the existence and continual development of multiple embedding models that produce incompatible embeddings. This often necessitates re-running a new embedding model (re-indexing) on all data after a model update, which can be costly. With the proposed Embedding Converter—a lightweight model—an already indexed dataset can be inexpensively transformed to be compatible with a target model’s embeddings. This approach can also facilitate deploying two models (a large model to index the corpus, and a lightweight model + embedding converter for real-time querying). Empirical results demonstrate the effectiveness of the proposed method for both in-domain and out-of-domain data across various model update scenarios.

**Strengths:**

- The paper is well-written and mostly easy to follow.
- The motivation and setup are reasonable and valuable, particularly for practitioners dealing with embedding models and large-scale retrieval systems.
- Empirical results demonstrate the efficacy of the proposed method on different datasets, tasks, and model update scenarios.
- The use case for latency reduction (using a light model + embedding converter) is a practical and useful proposal.

**Weaknesses:**

- While the motivation, method, and experimental setup in the paper are sound, the paper unfortunately overlooks an important and relevant domain of related work: model compatibility. The exact motivation, problem setup, and solution discussed in the paper have been the focus of model compatibility studies (e.g., see [1-4]). In particular, both [3] and [4] propose learning a simple MLP-based transformation function from the embedding space of a source model to that of a target model, and they address additional aspects (e.g., the availability of extra information, the potential for partial reindexing of the corpus with the new model) that are not covered in this work. All relevant model compatibility literature, including the mentioned studies, is absent from the current manuscript and should be discussed. The contribution of the proposed method, compared to the existing works cited above, is limited to addressing new domains/tasks.

- Section 3.2 (Loss functions):
  - For the relation-based losses, such as $L_{global}$ in equation (2), do we consider all pairs $(t_1, t_2)$ in the training dataset? Since the number of pairs scales quadratically, this approach is computationally expensive. Further clarification on this point—specifically, whether pairs are formed only within a batch (and, if so, the effect of batch size)—should be discussed.
  - In the analysis of loss functions, relevant works from the knowledge distillation literature should be reviewed. For instance, [5] proposes losses to capture relations between features, similar to what is proposed in this paper.

- One missing empirical analysis for the proposed method, also discussed in model compatibility literature, is the setup with multiple updates. All results provided in the paper consider a single update from source to target. The performance of the embedding converter should be discussed in scenarios with a chain of updates: $v1 \rightarrow v2 \rightarrow v3 \rightarrow v4 \rightarrow \dots$ .

[1] Shen, Yantao, et al. "Towards backward-compatible representation learning." Proceedings of the IEEE/CVF Conference on Computer Vision and Pattern Recognition. 2020.

[2] Hu, Weihua, et al. "Learning backward compatible embeddings." Proceedings of the 28th ACM SIGKDD Conference on Knowledge Discovery and Data Mining. 2022.

[3] Ramanujan, Vivek, et al. "Forward compatible training for large-scale embedding retrieval systems." Proceedings of the IEEE/CVF Conference on Computer Vision and Pattern Recognition. 2022.

[4] Jaeckle, Florian, et al. "FastFill: Efficient Compatible Model Update." The Eleventh International Conference on Learning Representations.

[5] Park, Wonpyo, et al. "Relational knowledge distillation." Proceedings of the IEEE/CVF conference on computer vision and pattern recognition. 2019.

**Questions:**

- Lines 276-277: It is stated that “queries are consistently encoded using the target model (gecko004) across all conditions.” Is this true for all columns in Table 1? For example, when the corpus is indexed by the gecko003 model, can the query be encoded by gecko004 (as in the column in Table 1 with 0.5067 average performance)? Since the embeddings of these models are not comparable, clarification is needed.
- In Table 5: What is the definition of global/local distance? Are these essentially average values of the loss functions defined in equations (2) and (3)? It might be helpful to indicate in the table that “lower is better.”
- In Table 5: What does “baseline” refer to? Is it the use of an identity embedding converter?

---

> ### Author Response · Authors · 2024-11-20
> **Response to Reviewer q2HY's comments (1/2)**
>
> We appreciate you bringing these relevant related works to our attention. Your insights are valuable in helping us better position and contextualize our contributions. We are also grateful for your positive feedback regarding the practicality and clarity of our paper. Please find our detailed response below, addressing each of your concerns. We welcome any further comments and will be uploading a revised manuscript shortly incorporating your suggestions.
>
> **Answer 1:** Thank you for bringing these relevant publications to our attention. We appreciate your thorough review and agree that these works hold relevance to our research. We will incorporate them into the related works section of the revised manuscript and provide a detailed qualitative comparison with our proposed method.
>
> While acknowledging similarities, we would like to highlight key distinctions between our approach and the cited works.  [1] and [2] propose methods (BCT) that require modifying the training process of the new model to ensure compatibility with the old model. This fundamentally differs from our setting, where both models are pre-trained and unavailable for further training. Consequently, BCT is inapplicable in our scenario, especially when converting between different embedding model families (e.g., gecko004 to OpenAI models).
>
> [3] introduces FCT, which addresses the limitations of BCT by employing converters (MLPs) to transform old embeddings into new embeddings, similar to our approach. However, FCT relies on "side information" unavailable in our context, limiting its practical applicability. While [4] eliminates the need for side information, its primary focus is on optimal online backfilling for improved conversion performance, a distinct objective from ours. Furthermore, [4] requires both embedding pairs and (image, label) pairs, which are unnecessary in our setting. It is also worth noting that [1]-[4] primarily focus on the image domain, while our method demonstrates broader applicability.
>
> The revised manuscript will include a comprehensive discussion of these related works, clearly delineating their differences and emphasizing the unique contributions of our proposed method.
>
> **Answer 2:** The reviewer correctly identifies the crucial role of batch training in ensuring scalability for our method. As detailed in Appendix B, we employ a batch size of 64 for each of the 13 datasets, resulting in a total batch size of 832.  Furthermore, to enhance scalability and manage computational costs, pairs are formed exclusively within each batch. This strategy allows us to effectively handle large datasets while maintaining training efficiency.
>
> **Answer 3:** Thank you for bringing reference [5] to our attention. We acknowledge the similarities between our embedding-converter and knowledge distillation methodologies, particularly regarding the use of distance and angle-based loss functions. While our objectives differ, we recognize the relevance of [5], specifically its exploration of distance-wise and angle-wise distillation losses, which share similarities with the global distance loss employed in our work but still distinct from our local distance loss.
>
> We will incorporate a discussion of [5] in the revised manuscript's related works section. However, we would like to emphasize that the primary contribution of our paper lies not in the specific loss function used, but rather in the introduction of a novel framework that significantly simplifies the practical migration of embeddings. This framework addresses a critical challenge in real-world applications and offers a valuable solution for facilitating embedding transfer.
>
> **Answer 4:** The reviewer raises a compelling point about the dynamic nature of real-world embedding scenarios. To address this, the revised manuscript includes an extended analysis encompassing the following evolutionary scenario: v1 (gecko002) -> v2 (gecko003) -> v3 (gecko004). This demonstrates the adaptability of our embedding converter to successive model updates.
>
> Furthermore, we explore a hybrid scenario where the corpus contains a mixture of converted embeddings and native target embeddings. This reflects practical situations where new data is gathered after a model update, allowing users to directly utilize the latest embedding model (partial backfilling). Our results demonstrate that the Embedding-converter effectively handles both scenarios, maintaining robust performance even with mixed embedding sources.

---

> > ### Author Response · Authors · 2024-11-20
> > **Response to Reviewer q2HY's comments (2/2)**
> >
> > **Answer 5:** To clarify any potential misunderstandings regarding our evaluation procedure, we have revised the manuscript to explicitly outline the following steps: (i) Source Model Performance: Both query and corpus embeddings are derived using the source model. (ii) Target Model Performance: Both query and corpus embeddings are derived using the target model. (iii) Embedding-Converter Performance: We utilize query embeddings from the target model. Corpus embeddings are generated using the source model and then transformed using the embedding converter. Therefore, our baseline (source model performance) employs the source model consistently for both query and corpus embeddings. This ensures a fair comparison and accurately reflects the impact of our embedding converter.
> >
> > It is important to note that directly combining embeddings from different models, such as using gecko004 for queries and gecko003 for the corpus, results in severely degraded performance (nDCG@10 < 0.01). This highlights the inherent incompatibility between different embedding spaces and underscores the need for an effective conversion mechanism like our proposed embedding converter.
> >
> > **Answer 6:** The reviewer is correct. The definitions of global and local distance are indeed consistent with Equations (2) and (3) presented in the paper. To address computational constraints, we employ subsampling of the corpus (using 1K samples) when calculating these metrics. This clarification, along with an explicit statement that lower distance values are preferable, has been added to the revised manuscript.
> >
> > **Answer 7:** To address potential confusion, we want to clarify the baseline used in our evaluation. The baseline performance represents the results achieved using the source model. We can compute distances between samples using the source model and compare with the ground truth which is derived from target model similarities. We will reword the caption for better clarity.
> >
> > Therefore, the evaluation focuses on comparing the source model's performance against the performance achieved when using converted embeddings generated by our embedding converter from the source model. This allows us to directly assess the effectiveness of the embedding converter in bridging the gap between the source and target embedding spaces.

---

> > > ### Comment · Reviewer_q2HY · 2024-11-23
> > > **Thank you for detailed response**
> > >
> > > The reviewer thanks the authors for their detailed response, which has addressed points of confusion. I would like to further discuss the contribution of the submitted work in relation to existing literature on Model Compatibility.
> > >
> > > The reviewer acknowledges that while the referenced papers primarily address vision models, the core problem remains the same—with replacing image embeddings with text embeddings. This reduces the novelty of the submitted method. However, the results are still valuable, given the wide applicability of text embedding models.
> > >
> > > Regarding the distinction with [3]: while [3] incorporates side-information, it also presents results without it, similar to the submitted work.
> > >
> > > In essence, the paper’s contribution is evaluated as applying an existing method (embedding conversion using a learned MLP for model compatibility) to a new domain (text embeddings) along with additional analysis. While the results are valuable, this reviewer still finds the contribution insufficient as the core problem/solution is not new. I would be happy to increase my initial score to 5 in acknowledgement of improvements during the rebuttal period.

---

> > > > ### Author Response · Authors · 2024-11-24
> > > > **Thanks for the quick response!**
> > > >
> > > > We appreciate the reviewer for reconsidering their score in light of our response. We are pleased to have clarified the points that were previously unclear.
> > > >
> > > > We acknowledge the similarity in the problem setting between our work and the suggested papers, which is to improve compatibility between old and new embedding models even though the validations are done in quite different domains. While [3] does explore methods without side-information (as shown in Table 6 of their paper), their results demonstrate a significant performance drop compared to utilizing it.  Additionally, [3] does not incorporate our crucial distance losses that we utilize and demonstrate to be critical in Table 6.
> > > >
> > > > In the revised manuscript, we will provide a detailed discussion that clarifies the similarities and differences between our work and the suggested papers. We thank the reviewer for bringing these relevant works to our attention, as they help us to more accurately position our contribution within the existing literature.

---

### Official Review · Reviewer_hZEJ · 2024-11-05

**Soundness:** 2
**Presentation:** 2
**Contribution:** 2
**Rating:** 6
**Confidence:** 3

**Summary:**

This paper proposes to translate the text embeddings of different embedding models, such as gecko003 of Google to openai-3-small. To facilitate training of such a translator, the authors propose a few loss functions: 1. regression loss to force the translated source embedding to be close to the target embedding, 2. global similarity loss, to retain the similarities of the source space in the translated space, 3. local similarity loss, which is similar to global similarity loss, but is done in a neighborhood of a selected piece of text t. The authors verified that the translated embeddings usually perform better than source embeddings, but slightly worse than target embeddings.

**Strengths:**

1. This work explores an interesting direction of pursuing compatibility between different embedding models. Although cross-embedder retrieval itself is not so strongly motivating, I believe it's important to have such embedding translators that bridge different models.
2. The 3 loss functions proposed seem to be reasonable.

**Weaknesses:**

My major concerns are 1) a few important details are missing, and 2) issues on experiments:
1. Some important technical details are missing. For example, in the global and local similarity losses, what is the Dist() used? Is this cosine?
2. How to get the k-nearest neighbors of a text piece t, $NN_k(t)$? Do we need to iterate the corpus and find similar embeddings? Could we simply perturb the embeddings of t to get its neighbors?
3. What are the number of parameters of the embedding converter?

Experimental issues:

4. Different embedding models should have drastically different topological structures of embedding spaces. Therefore, it would pose a severe challenge for such embedding translators. Although the authors have evaluated in experiments that Embedding-Converter performs well between source and target embedders, the evaluated embedding models are basically only 2 types: gecko (003 and 004, which I presume should be topologically homogeneous to each other) and openai-3-small. It's more convincing if more embedding models are evaluated to show that the architecture of Embedding-Converter is valid for other challenging pairs of embedders.
5. The reported retrieval performance of openai-3-small seems to be lower than reported in [a]. In Fig. 8(b) of [a], openai-3-small seems to score around 0.537 (no numerical values are reported, but on the chart it's well centered between 0.525 and 0.55), however Table 1 of this paper reports 0.5209.
6. Matryoshka-Adaptor [a] is not compared as a baseline, which is quite relevant. EDIT: I noticed that due to the review policy, arxiv papers released after 1st July don't have to be compared. However, if authors could make such comparisons, it would be a strong boost to the convincingness of this paper.

[a] Matryoshka-Adaptor: Unsupervised and Supervised Tuning for Smaller Embedding Dimensions.

**Questions:**

N/A

---

> ### Author Response · Authors · 2024-11-18
> **Response to the Reviewer hZEJ's comments**
>
> We appreciate the detailed feedback provided in your review. Our responses to your comments and questions can be found below.  A revised manuscript will be submitted shortly. We welcome any further comments or questions you may have.
>
> **Answer 1:** The reviewer is correct. We utilize 1-cosine similarity as our distance metric. This has been clarified in the revised manuscript to avoid any ambiguity.
>
> **Answer 2:** The reviewer raises a valid point about potential alternatives to our nearest neighbor selection strategy in Equation (3). During training, we leverage the known ground truth (target embedding similarities) and compute cosine similarity between target embeddings to efficiently identify the top-k nearest neighbors within a random corpus batch of 50k samples.
>
> While perturbing embeddings is an interesting suggestion, it's not directly applicable to Equation (3). This is because our formulation relies on the availability of corresponding source embeddings, which would be absent for perturbed target embeddings. Therefore, perturbation techniques are not suitable in this specific context.
>
> **Answer 3:** Regarding the architecture of our embedding converter, we employ a 4-layer multilayer perceptron (MLP) with hidden state dimensions adapted to the specific source and target embedding dimensions, as detailed in Appendix B. This adaptability is crucial to effectively bridge the gap between different embedding spaces.  For instance, when converting between gecko003 and gecko004 embeddings, the Embedding-converter comprises 35 million parameters. This highlights how low overhead the Embedding-converter is compared to the embedding model.
>
> **Answer 4:** The reviewer's point about the potential influence of topological similarity between embedding models is well-taken. While it's true that gecko003 and gecko004 likely share some degree of topological similarity, our goal was to demonstrate the versatility of the embedding converter across a broader range of scenarios. This includes cases where: (i) Models are similar but versions differ: As exemplified by gecko003 and gecko004, where the underlying architecture is related. (ii) Models are fundamentally different: Such as between gecko and OpenAI embeddings, which exhibit distinct architectural choices and even differing dimensionality.
>
> To further strengthen this claim, the revised manuscript incorporates results from experiments with other embedding models. These models are specifically chosen to possess no topological similarity with either gecko or OpenAI, providing a robust test of the embedding converter's capabilities.
>
> **Answer 5:** We appreciate the reviewer's diligence in comparing our results with those reported in [a]. It's important to highlight a key distinction between the two studies: [a] focuses on a subset of 8 BEIR datasets, while our evaluation encompasses the full set of 13 BEIR datasets. This difference in scope is explicitly stated in the caption of Figure 8(b) in [a], where the authors mention their use of 8 BEIR datasets for openai-3-small. Therefore, a direct comparison of performance numbers may not be entirely appropriate due to the differing evaluation settings.
>
> **Answer 6:** While we acknowledge the architectural similarities between our embedding converter and the Matryoshka-Adaptor, it's crucial to emphasize the fundamental difference in their objectives. Matryoshka-Adaptor focuses on enhancing the inherent "Matryoshka" properties within a single embedding model, aiming to improve its performance across various tasks. In contrast, our embedding converter tackles the challenge of translating between two distinct embedding spaces, effectively enabling interoperability between different models. Note that the Matryoshka-Adaptor method doesn't take the target embedding model into account during its process.
>
> Given these divergent goals, we believe that Matryoshka-Adaptor is not directly applicable to the embedding conversion problem addressed in our work.  The core functionalities and optimization strategies employed by each method are tailored to their specific objectives, making them distinct approaches despite sharing a similar architectural foundation.

---

> > ### Comment · Reviewer_hZEJ · 2024-11-25
> > **Thanks for the response**
> >
> > Your response addressed most of my concerns, except extra embedding models mentioned in Answer 4, since an updated version hasn't been uploaded yet. However, I'd like to raise my rating to weak accept before seeing the updated version. I do believe this is an interesting line of research, despite possible limitations on efficiency pointed out by Reviewer dLB8.

---

### Official Review · Reviewer_dLB8 · 2024-11-07

**Soundness:** 2
**Presentation:** 3
**Contribution:** 3
**Rating:** 6
**Confidence:** 3

**Summary:**

The authors focus on a specific use case of using embeddings generated from a pretrained LLM and transfer them to different target LLM. Typically, doing this would need reembedding the entire corpus using the target model. This paper present a way to eschew this re-embedding process by introducing a small converter (another Neural network) to learn mappings from the source model embedding space to the target model’s embedding space. Re-training on large internet-scale datasets requires an extensive amount of compute resources and time, and for production-scale systems that need to regularly update their models to SOTA. This can be an expensive process from both an effort and cost standpoint. The authors introduce an intermediate network (referred to as a converter in the manuscript) to learn the mapping between the embedding spaces of the models. They train this network by optimizing for similar pair-wise distances between points in their respective embedding spaces.

**Strengths:**

* The use case the paper tackles is interesting and relevant in the context of the growing use of pre-trained LLMs. The solution proposed is simple to understand and implement.

* The authors try to make their contributions to the setup quite generic and extensible to various models in the wild.

* The computation speedups achieved over the baselines are substantial, making this use case and strategy worth exploring.

**Weaknesses:**

* Even though the solution proposed does not require to re embedding with the target models, it still requires training a converter (and subsequently evaluating it/ performing hyperparameter searches) every time a new target model is introduced. In most practical scenarios, the datasets change across time just as much if not more than the embedding models. Changes to the dataset would need re-embedding and thus the proposed solution might have lower utility.

* Relevance of Relative Scores : It is not clear if the performance gains using the converter are actually reflective of the target model embeddings being better than the source model. For instance, in both in-domain and out-of-domain experiments presented, there are datasets for which the target embedding performance is better than the source embedding performance, however the converter embeddings have an even worse performance (for instance see DBPedia openaismall -> gecko004 in  Table 1), not indicative of the fact that the target embeddings are actually better than the source. This raises questions on how well do the converted embeddings reflect the relative performance of the source and target models - “It offers a preliminary performance guarantee when migrating to a new embedding model” - Sec 4.4. Furthermore, there are very few experiments where the retrieval performance of the source model embeddings is superior to that of the target (perhaps due to reasons mentioned in the next point), making it more difficult to assess the validity of the aforementioned claim.

* Fair Baseline Comparisons? Performance is measured using query embeddings generated using the target model across all scenarios - “queries are consistently encoded using the target model across all conditions” - Sec 4.2. This does not sound like a fair comparison to me - using target model query embeddings to evaluate the source model performance which uses a different embedding scheme altogether.
For  2 in-domain datasets, retrieval performance of the converter-embeddings is even better than the target embedding performance. Some explanation or insights into this behavior would be interesting perhaps.

* Potential Bias : Model selection and hyperparameter tuning was done based on retrieval performance rather than something related to the criterion which the converter was trained to maximize. Some details on how the validation set was constructed would be helpful here.

* No confidence intervals reported - (can use multiple seeds to initialize the converter model weights or use bootstrapping and provide error bars).

* Additional hyperparameter details like values of alpha and beta for some source-target model pairs could have provided additional insights to the reader.

* Not clear if the global loss in Eqn 2 is applied to all NC2 pairs in the training set,

**Questions:**

* A fairer comparison to baseline models?

* It would be informative to report error bars to note significance of results obtained, either a theoretical justification or at least a more thorough empirical evaluation of relative scores to establish their validity - for instance including more scenarios where the source model has a better performance than the target model and showing the claim on relative scores still holds.

---

> ### Author Response · Authors · 2024-11-18
> **Response to the Reviewer dLB8's comments (1/2)**
>
> We appreciate your thorough review and your acknowledgment of the method's practicality.  This rebuttal provides responses to your comments, and a revised manuscript with updated results will be submitted soon.  Please do not hesitate to raise any further points for clarification.
>
> **Answer 1:** Thank you for raising this important point. We acknowledge that new models would require retraining the embedding converter. This paper advocates for a streamlined workflow where model developers provide a pre-trained converter alongside any new model. This offers two key advantages: (i) Ease of use: Users can seamlessly experiment with new models without the burden of re-embedding their existing data. (ii) Efficiency: In dynamic environments with evolving datasets, only modified data points require re-embedding with the new model. Our proposed converter efficiently handles the remaining data without re-embedding, minimizing computational overhead.
>
> To further validate this approach, we conducted new experiments incorporating both re-embedded and converted data. The results demonstrate that this combined approach achieves comparable or even superior performance to relying solely on converted data. This underscores the effectiveness and practicality of our proposed embedding conversion framework, especially for large and evolving datasets.
>
> **Answer 2:** You correctly point out that performance gains from our Embedding-converter are not universally guaranteed across all datasets and model transitions.  However, our results demonstrate a strong trend: when transitioning to a superior target model, the Embedding-converter frequently enhances performance. Specifically, we observe improvements in 87% of the cases (20 out of 23) where the target model outperforms the source model on its own. This highlights the effectiveness of our approach in estimating the performance of target models in an efficient manner.
>
> While our primary focus is facilitating migration to newer, higher-performing models, we acknowledge the importance of evaluating the converter's performance in the reverse scenario (newer to older models). We have included these experiments in this rebuttal to provide a comprehensive analysis of the Embedding-converter's capabilities.
>
> **Answer 3:** We appreciate the opportunity to clarify our evaluation procedure, which has been detailed in the revised manuscript. To reiterate: (i) Source model performance: We used the source model for both query and corpus embeddings. (ii) Target model performance: We used the target model for both query and corpus embeddings. (iii) Embedding-converter performance: We used the target model for query embedding and the converted source model embedding for corpus embedding.
>
> This methodology ensures a fair comparison. Using mismatched embeddings (e.g., gecko004 for queries and gecko003 for the corpus) results in negligible performance (nDCG@10 < 0.01) due to inherent incompatibilities, further emphasized by the differing dimensionality of gecko and openai-small embeddings.
>
> Regarding the few instances where the Embedding-converter slightly outperformed the target model, we attribute this to experimental variance. This occurred in only 2 out of 26 datasets (<10%).  To address this, we have included an analysis of performance variance in the rebuttal, which supports our conclusion that these minor differences are statistically insignificant.
>
> **Answer 4:** This response addresses concerns about potential bias in our validation process. We utilized the Scifact dataset (comprising 1109 queries, 1258 labels, and 5183 corpus passages) for validation. While significantly smaller than the full BEIR training dataset, its retrieval performance served as the basis for hyperparameter optimization and model selection. Importantly, our experiments revealed that performance was relatively insensitive to both hyperparameter values and the specific validation set used.  Most hyperparameters were fixed (see Appendix B), with only alpha and beta subject to optimization. Even then, we explored a limited grid search of (0.01, 0.1, 1.0) for both parameters. Therefore, we argue that the results presented are robust and not unduly influenced by the choice of data, task, or hyperparameters.
>
> **Answer 5:** We appreciate the reviewer's suggestion. To address concerns about the robustness of our results, the revised manuscript now includes an analysis of the variance observed across experiments conducted with different random seeds for test set selection. This provides further evidence of the stability and generalizability of our findings.

---

> > ### Author Response · Authors · 2024-11-18
> > **Response to the Reviewer dLB8's comments (2/2)**
> >
> > **Answer 6:** The optimal values for alpha and beta were determined based on performance on the validation set.  Empirically, we found that setting alpha = beta = 0.1 consistently yielded strong results across all datasets. This finding has been incorporated into the updated manuscript. All other hyperparameters remained fixed throughout our experiments.
> >
> > **Answer 7:** The reviewer is correct to point out the importance of scalability for large corpus.  We confirm that batch training was employed for all three loss functions to ensure computational efficiency. Details regarding batch sizes are provided in Appendix B.

---

> ### Comment · Reviewer_dLB8 · 2024-11-27
>
> I thank the authors for addressing my questions/concerns well. While the line of research is exciting and worthy of discussion, after reading other reviewer's comments, I'm still unconvinced about their proposed solution's utility. The authors, indeed, have addressed most of my other concerns. I'm happy to increase my score accordingly.

---

### Author Response · Authors · 2024-11-26
**Revised Manuscript Addressing Reviewer Comments**

We thank the reviewer for their insightful and impactful comments. We greatly appreciate the time and effort invested in reviewing our paper. We have carefully considered your feedback and updated the manuscript accordingly (in red color). Key revisions include:

- **Expanded experiments**: To demonstrate the generalizability of Embedding-converter, we have included new experiments in the Appendix covering:

(i) Conversion between different embedding model versions (both new-to-old and across multiple versions),

(ii) Conversion between models with different dimensions (smaller to larger).

(iii) Conversion from open-source to black-box models.

(iv) Evaluation of Embedding-converter with mixed corpus embeddings.

- **Clarified hyperparameters and methodology**: We have provided detailed information on hyperparameter settings (including ranges and defaults), validation set details, and the number of parameters in Embedding-converter. The manuscript now also includes a clearer explanation of the distance definition and batch training process.
- **Enhanced experimental descriptions**: We have clarified the metrics and experimental settings used to evaluate Embedding-converter.
- **Revised related works**: We have incorporated the suggested related works and revised the section to better position our contributions.

The revised manuscript is available for your review. We welcome any further comments or questions you may have.

---

### Meta-Review · Area_Chair_iLJD · 2024-12-19

**Metareview:**

This paper introduces a domain-general method to convert embeddings from a source model to a target model, even if the dimensionality of the embeddings differ, without necessitating retraining or modifying the pre-training process. The crux of the approach involves formulating a loss function that can handle different dimensionality embedding spaces, similar to the philosophy behind dimensionality reduction techniques like stochastic neighbor embedding. Also, similar to dimensionality reduction/visualization techniques, the authors introduce both global and local similarity measures that can be balanced via multitask losses. The method is validated through experiments on language tasks both within and across model families, and is also evaluated on OOD tasks.  The authors advocate for leveraging their embedding converter framework in model releases, where both models and converters are released together, facilitating model comparison and adoption without the need for expensive retraining or embedding production.

Strengths:
Simple, practical approach to converting embeddings, with efficient implementation (no need for fine-tuning, etc.) Also nice to see a consideration of an approach to handle both within and across model types/embedding dimensions. Well written paper in general. Experiments consider a broad range of use cases.

Weaknesses:
More info on the technical aspects of the embedding converter method would be helpful in the intro as it comes across as a blackbox until reading through Methods section. It would be good to also review connections to dimensionality reduction given similar loss formulations. While the method is potentially of practical value in certain settings like model comparisons in a fast moving field, it has somewhat limited utility given that datasets also change quickly. Previous literature on model compatiblity overlaps with the goals of this paper, and the method uses loss formulations that have been explored for model distillation and dimensionality reduction; this softens the novelty factor, making the contribution more oriented towards practical settings. While the experimental results are pretty comprehensive and were enhanced by the rebuttal, there is still limited insight and analysis into why this method is particularly effective, and why the regression loss is not sufficient but the tailored mult-objective loss works better (the ablations are useful though!)


Decision reasoning:
I want to thank both the authors and most of the reviewers for engaging in dialogue and helping to enhance the. paper. I found it interesting and of potential practical value. However, I think more detailed analyses should be conducted to understand why this method seems to work well over simpler methods that do not include the global and local similarity losses, and why previous methods on model compatibility would not work as well. Since reviewers are generally borderline (even disregarding the reviewers who did not engage) I am inclined to suggest that the authors take the opportunity to enhance the intro and analysis to provide more useful insights to the ML community to generate more excitement about the method.

**Additional Comments On Reviewer Discussion:**

The first three reviewers (dLB8, hZEJ, and q2HY) were very helpful in pointing out weaknesses of the original submission and engaged with the reviewers to enhance the paper significantly. Their ratings remain in borderline territory, however, mostly due to concerns over the realistic utility of the method and remaining skepticism about limited novelty relative to existing methods.
Unfortunately, the fourth reviewer (MnsG) is not an expert in the area and did not provide useful additional value (and simply agreed with other reviewers.
The fifth reviewer (h1G5) provided some useful feedback but did not engage in discussion, so it’s not clear to me whether they would have raised their scores based on author responses. My sense is that the authors generally addressed the reviewer concerns, but I agree that there is still a lack of interesting analyses/insights about the method that would make the paper more suitable for this conference — over it’s potential (but possibly limited) practical value.

---

### Decision · Program_Chairs · 2025-01-22

Reject